# Automatic heliostat learning for in situ concentrating solar power plant metrology with differentiable ray tracing

Max Pargmann [1,6] ✉, Jan Ebert [2,3,6], Markus Götz [2,4], Daniel Maldonado Quinto [1], Robert Pitz-Paal[1,5] & Stefan Kesselheim [2,3]

Concentrating solar power plants are a clean energy source capable of competitive electricity generation even during night time, as well as the production of carbon-neutral fuels, offering a complementary role alongside photovoltaic plants. In these power plants, thousands of mirrors (heliostats) redirect sunlight onto a receiver, potentially generating temperatures exceeding 1000°C. Practically, such efficient temperatures are never attained. Several unknown, yet operationally crucial parameters, e.g., misalignment in sun-tracking and surface deformations can cause dangerous temperature spikes, necessitating high safety margins. For competitive levelized cost of energy and large-scale deployment, in-situ error measurements are an essential, yet unattained factor. To tackle this, we introduce a differentiable ray tracing machine learning approach that can derive the irradiance distribution of heliostats in a data-driven manner from a small number of calibration images already collected in most solar towers. By applying gradient-based optimization and a learning non-uniform rational B-spline heliostat model, our approach is able to determine sub-millimeter imperfections in a real-world setting and predict heliostat-specific irradiance profiles, exceeding the precision of the state-of-the-art and establishing full automatization. The new optimization pipeline enables concurrent training of physical and data-driven models, representing a pioneering effort in unifying both paradigms for concentrating solar power plants and can be a blueprint for other domains.

Concentrating solar thermal power plants (CSPs) are an essential part of the ongoing energy transition[1,2]. They are not only able to provide dispatchable electricity, but also direct heat for industrial processes or the synthesis of carbon-neutral fuels[3–6]. CSPs particularly stand out due to their power conversion efficiency, competitive levelized cost of energy, low consumption of rare materials compared to photovoltaics, and their ability to store generated energy for several days[7–9]. In typical setups, thousands to hundred thousands of mirrors, the *heliostats*, reflect sunlight onto an absorbing surface, the *receiver*. The solar radiation from the superposition of the individual heliostat focal spots generates thermal power, reaching radiant fluxes of 400 MW and temperatures exceeding 1000 °C.

Thereby, the heliostat field is one of the largest contributors to the energy yield as well as levelized cost. Heliostat technology is consequently subject to high research interest and rapid engineering improvements[10]. High-quality optical heliostat performance at minimal cost is a key design goal to achieve CSPs' commercial success. For example, it is estimated that a reduction of the glass thickness from

[1]Institute of Solar Research, German Aerospace Center (DLR), Köln, Germany. [2]Helmholtz AI, Köln, Germany. [3]Jülich Supercomputing Centre, Research Institute Jülich (FZJ), Jülich, Germany. [4]Scientific Computing Centre (SCC), Karlsruhe Institute of Technology (KIT), Karlsruhe, Germany. [5]Chair of Solar Technology, RWTH Aachen University, Aachen, Germany. [6]These authors contributed equally: Max Pargmann, Jan Ebert. ✉e-mail: max.pargmann@dlr.de

4 mm to 3 mm can reduce field cost by 5%, summing up to an impressive $3 million for a full-size concentrating solar power plant[11].

While such cost reductions effectively decrease overall expenses, they simultaneously increase operational challenges. The tight cost constraints in heliostat manufacturing manifest in heliostat deficiencies such as surface deformations or cause sun tracking errors. The receiver is then likely to be exposed to thermal stresses and heat spikes, significantly reducing the components' longevity. Operators of CSPs are already forced to run lower temperatures than possible and thus reduced efficiencies. To safely reach higher radiant fluxes, the heliostats need to precisely focus on a pre-determined location at the receiver, while maintaining a specific power distribution. Each individual heliostat must hit the receiver at a predetermined aim point, which is dependent on the one hand on external conditions, such as the position of the sun, the alignment of the other heliostats, e.g., due to shading and blocking, or possible cloud passage[12]. On the other hand, internal conditions, such as alignment errors or mirror deformations, can lead to deviations in the irradiance in the aim point and focal spot shape. It is possible to improve the power distributions if known heliostat deficiencies are taken into account.

For example, heliostat misalignment can be corrected by so-called heliostat calibration. The most common method is the *camera-target* method. In this approach, suggested first by Baheti et al.[13], a single heliostat's solar reflection is diverted to a calibration target near the receiver. Due to the heliostat's inaccuracies, its aim point will differ from the designated one. This difference from one or more focal spot images can be used to optimize a geometric model of the heliostat. In most cases, this is an error-based alignment model minimizing the distance between the designated and the actual aim point. Refined over several years, this approach is now the most commonly used method for heliostat alignment calibration[14–26]. While several calibration techniques have been proposed, with varying (dis-)advantages in terms of cost, accuracy or speed[27–30], the *camera-target* method is the de facto standard for CSPs across the world.

A heliostat's deformation, more precisely its slope error, can for example be determined by stripe pattern deflectometry[31]. In this widely used technique[32–36], a stripe pattern is projected onto a reflective object and the resulting projection is measured. From the distortions of the stripe pattern, the surface of the object can be reconstructed. While this method leads to precise results under laboratory conditions, the measurement of heliostats directly in the heliostat field is difficult. At the solar tower, this process is performed at night. Striped light patterns are projected onto the calibration target and a camera detects the reflected pattern. Although the accuracy is still high, the reliability of this method is greatly reduced in environmental conditions found at actual solar towers. Besides inherent instabilities like quantization of the intensity, non-linearity of the detector, unwanted background variations and other types of optical and electronic noise[37], this task is particularly challenging due to dew, dust, long exposure times, misplaced or misaligned heliostats, and wind. Hence, it has not experienced widespread use since its first publication in 2011[38]. Other methods for obtaining the heliostat shape, e.g., photogrammetry, tend not to be as easily implementable or not as cost effective in the heliostat field compared to deflectometry[39].

For optimal power plant control, the irradiance is the most important input variable. With accurate knowledge of the heliostat field, heliostat geometry and heliostat surface, it can be predicted from simulations for a given distribution of aim points on the receiver. The knowledge about the irradiance allows for modern aim point distribution[12,15,40,41], accurate model predictive control[42–44], and reduced safety margins in plant operation. For example, only rudimentary improvements of the heliostat geometry model have lead to power gains of 20% when adjusting the aim point distribution strategy[41]. Hence, considerable efforts are made to support aim point optimization algorithms with realistic heliostat irradiance profiles. The

approach by Sanchez et al.[45] identifies the canting errors of heliostats to reach a higher overlap of simulation and measurement. Zhu et al.[41] created a heliostat model in which four parameters per facet are fit to observational data. Although the obtained surface is simplistic, the improved irradiance prediction for distant heliostats led to the realization of the power gains mentioned above. The need for even better irradiance predictions is also the motivation behind the recent work of Martinez et al.[46]. They were able to show how surfaces can be reconstructed from focal spot images with an accuracy close to that of deflectometry measurements during day time. Their main limitation is that the target-heliostat distance is restricted to roughly 10 m. This distance is too small to use the existing infrastructure, so they have to resort on a target moving through the heliostat field.

Heliostat metrology is a key ingredient to establishing the wider adoption of CSPs. Despite the ongoing efforts, the *Roadmap to Advance Heliostat Technologies for Concentrating Solar-Thermal Power Plants* by the U.S. National Renewable Energy Lab identifies optomechanical error measurement in outdoor environments as a necessary but currently unattained key factor for large-scale commercial success[11]. Especially in-situ metrology methods, i.e., measurements without removing the heliostat from its mount, are desired as they allow for integrating such measurements into the operation of plants at low cost and with minimal operational adjustment.

To address these challenges we present a machine-learning solution facilitating the derivation of heliostat characteristics using the existing calibration metrology infrastructure. For this, our technique extends commonly used ray tracing approaches[47–54] with a differentiable model formulation driven by in-situ acquired data. This enables us to compute the derivative of the ray-traced pixel intensities with respect to all parameters affecting the light direction. In turn, we can formulate a supervised regression problem that infers the trainable input parameters by minimizing an objective function (loss) indicating the difference between the generated ray tracing image and the in-situ measured calibration data. As the corresponding heliostat surface model, we additionally propose the use of a differentiable and therefore learning non-uniform rational B-spline (NURBS) variant.

This publication concentrates on the reconstruction of surfaces from focal spots and the prediction of irradiance profiles. However, every heliostat parameter affecting the direction of the light, e.g., the geometric model for alignment correction, may be optimized analogously and even simultaneously to the surface reconstruction in the same manner. Using this method, we are able to determine submillimeter imperfections of nearly planar heliostats in a real-world setting and predict heliostat-specific irradiance profiles with high precision in all considered experimental cases. Contrary to the state-of-the-art approach, i.e., stripe pattern deflectometry, our technique can be used during regular power plant operation. To assess the effectiveness of our approach, we conducted a proof-of-concept field test at a research CSP facility in Jülich, Germany. Our method consistently enhances the annual irradiance forecast, achieving the same accuracy as simulations leveraging deflectometric measurements, enabling the implementation of modern optimal aim point strategies[55]. Even for smaller power plants, such as Gemasolar in Almeria, our approach has the potential to increase yearly revenues by up to 39%, equating to approximately $3 million[56]. The employed method therefore poses ametrology tool for in-situ heliostat measurement and operation optimization. In complementary simulations, we show that our approach generalizes to all heliostats of the whole array. A schematic overview of the process and the main results are depicted in Fig. 1.

## Results
### Ray tracing at concentrating solar power plants
Ray tracers have become an invaluable tool for CSPs[48,50,57–59]. For example, they are used in planning field layouts[60], the prediction of the

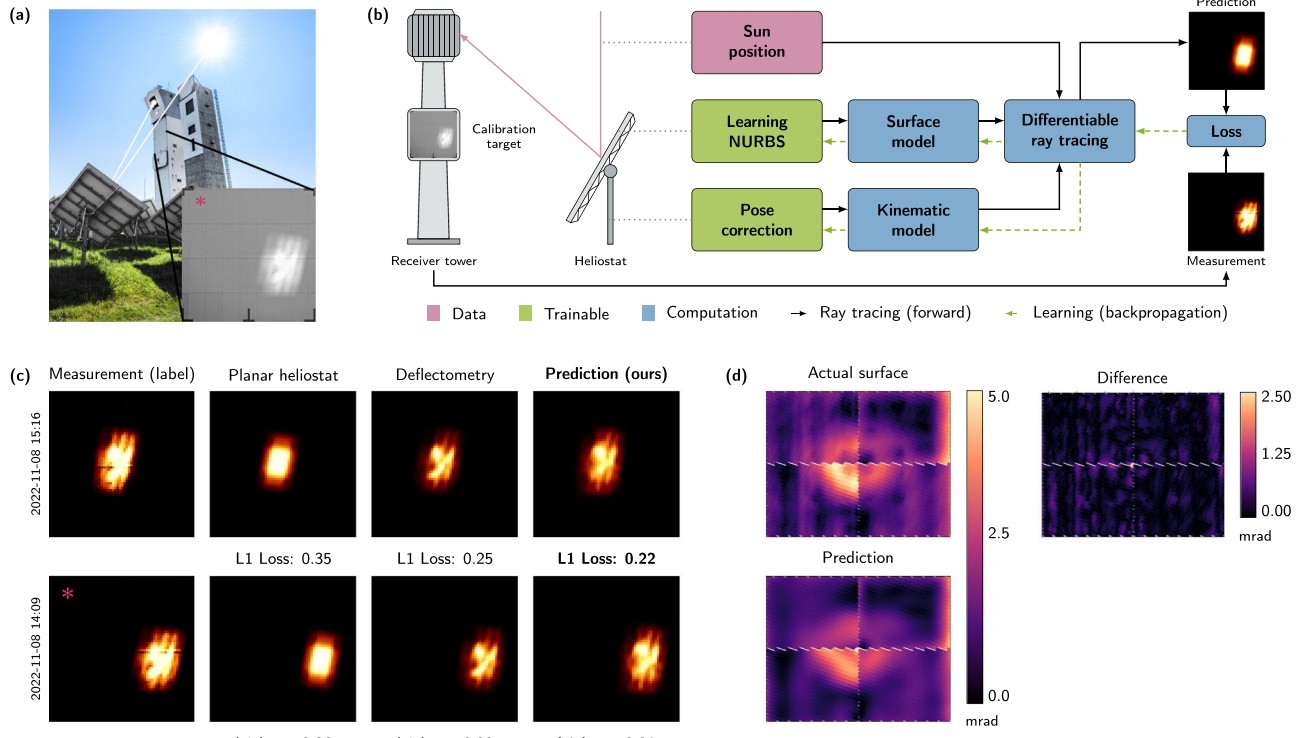

**Fig. 1 | Overview of the machine learning pipeline and results on data from the solar tower in Jülich. a** Image of a CSP in Jülich, Germany. The heliostats, shown from behind, focus the sunlight on the receiver surface of the tower. Located below the receiver is the calibration target. The inset (*) shows a focal spot image as taken during calibration. **b** Schematic overview of the in-situ optimization of CSPs with differentiable ray tracing able to learn the heliostat parameters. **c** Comparison of the irradiance profiles obtained from (left to right) measurement, calibration images with naive ray tracing of a planar heliostat, a supporting deflectometry measurement, and the prediction of the differentiable ray tracer. Deviation between the measurement and each generated image is quantified using the mean absolute error (L1 loss, ↓, lower is better). The image marked with * is the normalized calibration target from the inset above. **d** Heliostat surface reconstructions from calibration images using learning NURBS. The surfaces are represented by the deviation in mrad of their normals from an ideal planar surface. The heliostat was at 50 m distance from the receiver.

annual energy yields[61], the design of suitable structures for the synthesis of climate-neutral fuels[62], or flux density predictions[63]. On a physio-technical level, the ray tracing problem at CSPs combines specular reflection of the sunlight on the heliostat and diffuse reflection at the target. The diffuse reflection requires integration over all incident angles, corresponding to a high total number of rays. Therefore, ray tracing simulations typically employ Monte Carlo sampling[64] of the rays emitted by the sun proportional to the respective solar radiance. This *importance sampling*[65] technique allows for a drastic reduction of the required rays and is also the foundational mechanism used in our proposed method.

Consider now a single heliostat's reflected image moved onto a calibration target, creating an irradiance profile. The calibration target's surface is matte, i.e., a diffusely reflecting Lambertian surface[66]. Its reflected light is proportional to the surface irradiance, independent of an observer's viewpoint. The irradiance $E$ at position $\vec{x}$ on the calibration target can then be obtained by integrating the radiance $L$ over all incoming directions $\vec{t}$, multiplied by the cosine of the incident angle $\theta$. Neglecting ambient lighting, the incident irradiance can be constructed by finding the intersection $\vec{h}$ of the incident direction with the heliostat, and, if within the heliostat surface, constructing the reflected direction $\vec{t}_r$ by evaluating the local heliostat normal $\vec{n}_h$ and the solar radiance $L_\odot$ in the reflected direction:

$$E\left(\vec{x}\right) = \int_\Omega L_\odot\left(\vec{t}_r\left(\vec{t},\vec{h}\right)\right)\cos\theta\, d\vec{t}.\qquad(1)$$

$\Omega$ denotes the half-sphere of incident rays. This formulation permits the inclusion of terms to model real-world imperfections, e.g., a heliostat's surface deformation from an ideally planar one. We express this by introducing the reflectivity function $\vec{t}_r$ that depends on the heliostat surface point $\vec{h}$. Then, most ray directions will not contribute to the irradiance of a point on the target surface due to the small size of the solar disk. See Fig. 2a for a schematic visualization of the ray tracing process and the associated geometry.

## Differentiable ray tracing

Inspired by differentiable ray tracers in computer graphics, such as redner[67], RayTracer.jl[68] or Sionna[69], we reinterpret Eq. (1) as a mathematical function that maps from geometric input parameters to a matrix-valued output image. Computing the derivative, or more precisely the Jacobian, of the output image with respect to the input parameters then allows for solving a wide variety of inverse problems. In our case, we are interested in determining the geometric parameters, e.g., heliostat properties. Thus, we can formulate a supervised regression problem in which we infer the parameters or, in other words, the unknown weights of our machine learning model, through gradient-based minimization of a loss function fitting the ray-traced image as close as possible to observational data (cf. Fig. 1b). This allows us to not only predict heliostat alignments, but also their surface deformations or any other parameter with a differentiable contribution to the loss. While these other parameters are not further considered in this work, their exploration is of great interest for general CSP operation.

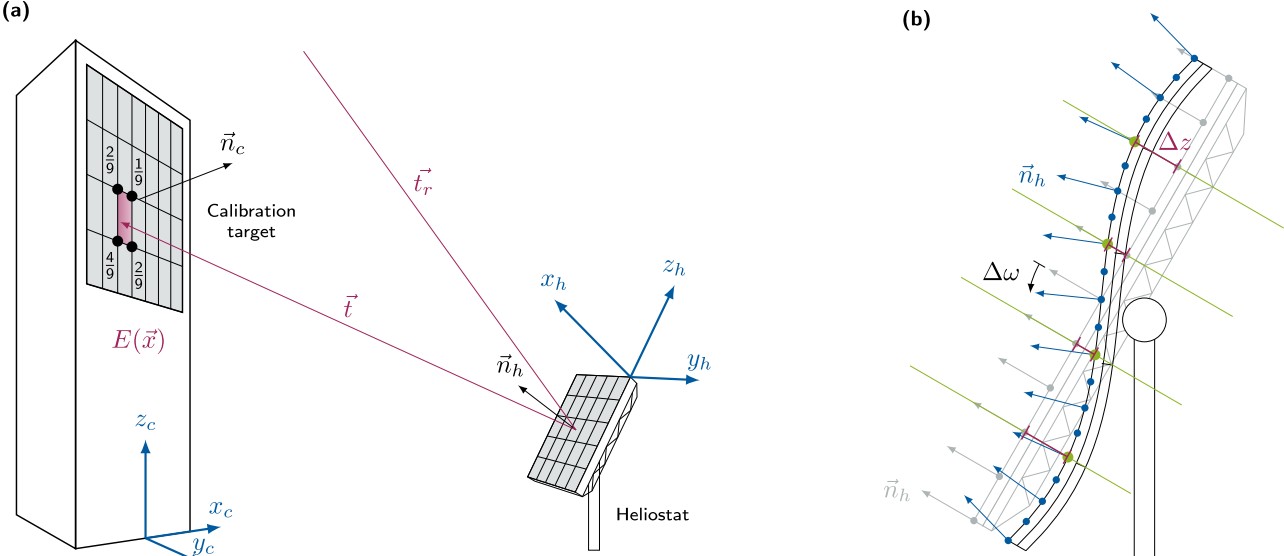

**Fig. 2 | Sketch of the differentiable ray tracing process using NURBS.**
**a** Schematic drawing of the ray tracing process and the binning function. Starting from a heliostat lattice point with normal $\vec{n}_h$, an incoming ray is reflected and traced ($\vec{t}$) to the target. Its intensity is linearly distributed to the $N = 4$ nearest lattice points (black circles). The different coordinate systems are shown in blue. **b** Schematic drawing of a heliostat NURBS surface. The control points $P$ (green

dots) are shifted in the $z$ direction (red section on green line) away from the ideal surface (gray). This deformation modulates the normal vectors $\vec{n}_h$ (blue). One discrete point is moved by $\Delta z$, which influences the reflected ray direction by the slope error $\Delta\omega$. The information about the infinitesimal change $\Delta\omega$ can be traced back via automatic differentiation to the change $\Delta z$ of the NURBS control points.

The parameter learning procedure is based on reverse-mode differentiation and visualized in Fig. 1b. All operations are first executed in the computational *forward* direction (solid black edges). For any path of operations that are connected to a trainable weight (green nodes), the gradient is subsequently determined in the *backward* direction (green dashed edges). This backward direction efficiently calculates the sequential gradient of the forward operations with respect to the optimization parameters via backpropagation[70,71]. This is realized by continual application of the chain rule from calculus to all nested functions to be differentiated. The loss propagation happens in reverse order, i.e., it starts with the calculated loss and ends at the learnable weights. It is important to emphasize that the backward direction needs to be executed exclusively for the computational pathways leading to trainable parameters. Notably, this implies that only these computations have to be differentiable.

Internally, we are building on the numerical theory of several current-generation ray tracers like STRAL[58,59]. This includes for example the reduction of the number of necessary reflection calculations by starting the ray tracing process on a discretized heliostat. For each of the heliostat points, rays are sent according to the local normal vector and the sun's position. As a result, a ray that is reflected more than once on the heliostat does not contribute to the receiver image and is not tracked further (see Fig. 2). One of the usually employed mechanisms is the so-called hard binning, i.e., the accumulation of rays based on their intersection point $\vec{x}$ of the nearest point on a grid overlaying the target. While computationally fast, the discretization of the rays is not differentiable. Therefore, we relax the formulation to a soft binning scheme, which distributes the rays linearly on the calibration target (compare Fig. 2a), a common anti-aliasing technique in computer graphics[64]. In other contexts, this is also known as a regularization of the delta function which conserves the first-moment of the ray-target-intersection[72]. For this, we introduce a differentiable weighting function $\gamma\left(\vec{x}_{ij}, \vec{x}\right)$ describing the irradiance contribution of a ray hitting the target at position $\vec{x}$ to the pixel centered at $\vec{x}_{ij}$. Then, the irradiance is a weighted sum over all rays cast from heliostat surface point $\vec{h}_l$ in direction $\vec{t}_k$. We define the intersection of the reflected ray with

the target as $\vec{x}(\vec{h}_l, \vec{t}_k)$ and write up to a prefactor:

$$E_{ij} \propto \sum_{\substack{\text{ray } k, \\ \text{position } 3l}} \underbrace{\gamma\left(\vec{x}_{ij}, \vec{x}(\vec{h}_l, \vec{t}_k)\right)}_{\gamma_{ijkl}} \cos\theta. \qquad (2)$$

The factor $\gamma_{ijkl}$ describes the contribution of ray $k$, reflected at discrete position $l$, to the discrete grid point (pixel) $\vec{x}_{ij}$.

We have implemented the differentiable ray tracer using the machine learning framework PyTorch[73]. All computations are composed from differentiable primitives permitting the evaluation of the Jacobian using PyTorch's automatic differentiation mechanism in backward mode. Learning, i.e., adjustment of the trainable weights, happens through application of the gradient via an optimization algorithm such as Adam[74]. A practical side effect of the PyTorch implementation is the option to accelerate computations with specialized hardware like GPUs.

## Learning NURBS

A key component of our method is a differentiable heliostat surface model. Physics requires this surface to be smooth. For this, we propose a variant of *non-uniform rational B-splines* (NURBS)[75]. NURBS are a mathematical representation of $n$-dimensional geometry as a linear combination of a B-spline curve basis. The NURBS control points directly correspond to the trainable parameters of our machine learning model (cf. Fig. 2b). This formulation does not only ensure differentiability and smoothness, but also provides the flexibility to represent arbitrary surfaces and their deformations with variable degree of detail based on the chosen polynomial degree of the NURBS. More specifically, the spline degree controls how many of its neighboring discrete points are affected by the modulation of another point. We employ the L1 loss, i.e., the mean absolute error, as optimization criterion due to its high sensitivity to deviations in low-intensity regions compared to the mean square error. We scale the loss with the number of pixels. This way, the computed losses are comparable even for different image resolutions.

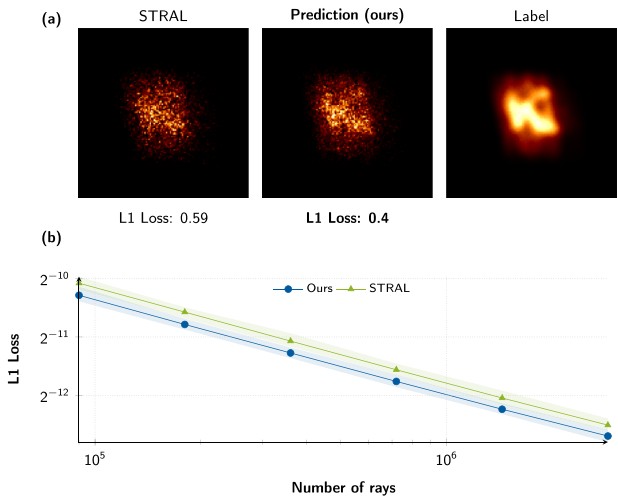

**Fig. 3 | Comparison to the state-of-the-art ray tracing tool STRAL. a** Illustration of the anti-aliasing property of the differentiable ray tracer compared to STRAL. Both approaches use a small number of rays (6000) in contrast to a reference, denoted *label*, with a large number of rays (1,500,000). Our image is visibly better resolved due to the differentiable soft-binning scheme, leading to a lower L1 loss (↓, lower is better). **b** Quantitative comparison of the image quality of our proposed differentiable ray tracer and STRAL measured with the L1 loss (↓, lower is better). Solid line depicts average values, transparent band the minimum and maximum values, out of 54 scenarios.

For such a NURBS model, we also assume that learned parameters with small deviations from an idealized planar heliostat surface are more realistic and therefore preferred. Thus, we would like to penalize large displacements of the NURBS control points from ideal positions through regularization. For example, employing a term proportional to the square of the distances of the NURBS control points in the normal direction $\Delta z$ of the heliostat requires the solution to employ the smallest values being compatible with the observed images, as found in ridge regression[76]. With a penalty term proportional to the absolute magnitude, following the idea of Lasso[77], only the set of necessary parameters will converge to a nonzero value, leading to only sparsely deformed heliostats. Depending on the application, the optimal regularization factors vary, and even high-order or additional terms may be beneficial. In our experiments we have found regularization not to be of importance for simple geometrical problems, such as surface reconstruction. Yet, for more complex geometrical arrangements, e.g., simultaneous surface-heliostat alignment, it is a meaningful tuning factor.

## Comparison with classical ray tracing

For our quantitative comparative analyses, we are confining us to an evaluation against STRAL[63]. This choice is premised upon (a) its image quality, which is able to predict heliostat irradiance profiles with an exceptionally high level of accuracy compared to real-world observations. This has been tested for example using highly resolved deflectometric surface profiles measured at the CESAR-1 solar tower power plant, achieving nearly 98% accuracy overlap to irradiance measurements[63]; (b) representative benchmarks for a wide set of classical ray tracers for solar towers[58,59] (c) its high computational speed[59]; making it an ideal choice to compare with our deflectometric simulations.

STRAL and our differentiable ray tracer produce the same irradiance images, as the number of traced rays tends towards infinity. However, in finite regimes, i.e., a low number of rays, the suggested differentiable soft binning scheme results in images with a finer level of detail with fewer rays while also converging faster. This effect is

illustrated in Fig. 3. Notably, this is only applicable for differentiable receiver geometry. Discontinuities, e.g., steps on the receiver surface, are challenging for our proposed approach.

With respect to computational performance, we observe a slight advantage in wall time when using the proposed differentiable ray tracer with an equivalent number of rays, despite the higher computational cost of binning. We want to mention that a fair comparison is complex and comes with several caveats. While STRAL is a highly optimized code that parallelizes over all physical simulation objects, it is only executable on CPUs. In contrast, the differentiable ray tracer runs unoptimized, parallel code only over the set of rays from a single heliostat, but is able to leverage acceleration from GPUs.

## Field test at a concentrating solar power plant

We have benchmarked our proposed ray tracing approach against data collected in a field test at a real-world concentrating solar power plant in Jülich, Germany. This research facility can generate rated electrical power of up to 1.5 MW by using over 2000 heliostats at a distance between 25 and 250 m. Each heliostat has four individual facets, which are *canted*, i.e., tilted inwards to achieve a joint focus, and which have an astigmatically corrected target alignment[78]. For the validation procedure, we selected a heliostat in the first row of the field at 25 m distance. As a comparative baseline, we measured the heliostat's surface using deflectometry on October 21st, 2021. Under clear sky conditions on March 4th, 2022, we additionally performed the regular calibration procedure for the same heliostat at two different times of the day. These calibration images correspond to our training data. During the next regular calibration interval, roughly eight months later and on different times of the day, two additional images were acquired representing our test data.

Figure 1c shows the obtained results on our held-out test data set. The measured ground truth images, i.e., the regression labels, can be found in the left-most column denoted as *measurement*. For training, the image's mean value was subtracted from each pixel and the images were intensity-normalized. The column *planar heliostat* displays the irradiance predicted by a ray tracing method that assumes an idealized heliostat without any surface deformations. This corresponds to the current implementation in most operational concentrating solar power plants. In the third column, flux density predictions obtained from ray tracing based on real-world *deflectometric* measurements are depicted. Finally, the right-most column shows the *predictions* of our proposed differentiable ray tracer. As expected for a real-world heliostat calibration image, the focal spot is off-center. This misalignment stems from mechanical errors of the heliostats in Jülich, which use a primary horizontal axis and a secondary perpendicular axis. This deviation is learned by our differentiable ray tracer through optimization of a geometric model by comparing the measured and the simulated orientation, initially ideally centered. However, for this publication, we do not use a realistic two-axis geometry model, but rotate the heliostat in the mirror origin around three axes. The rotations determined in this way must be determined individually for each image and cannot be used for heliostat calibration, but guarantee that each image has the maximum overlap with the measurement. To achieve better comparability, all non-data-driven approaches use the same heliostat alignment and rotation correction derived from our ray tracing pipeline. The color map image is normalized by the maximum incoming intensity and spatially scaled to match the calibration target dimensions.

It is apparent that both deflectometry and our proposed approach achieve qualitatively and quantitatively significantly better irradiance profiles compared to the naive assumption of a planar heliostat. The differentiable ray tracing does not only match deflectometry, but slightly improves on it. It is important to stress that the images shown here have *not* been used for training. Hence, using only the two training images obtained eight months prior (see the Training Data

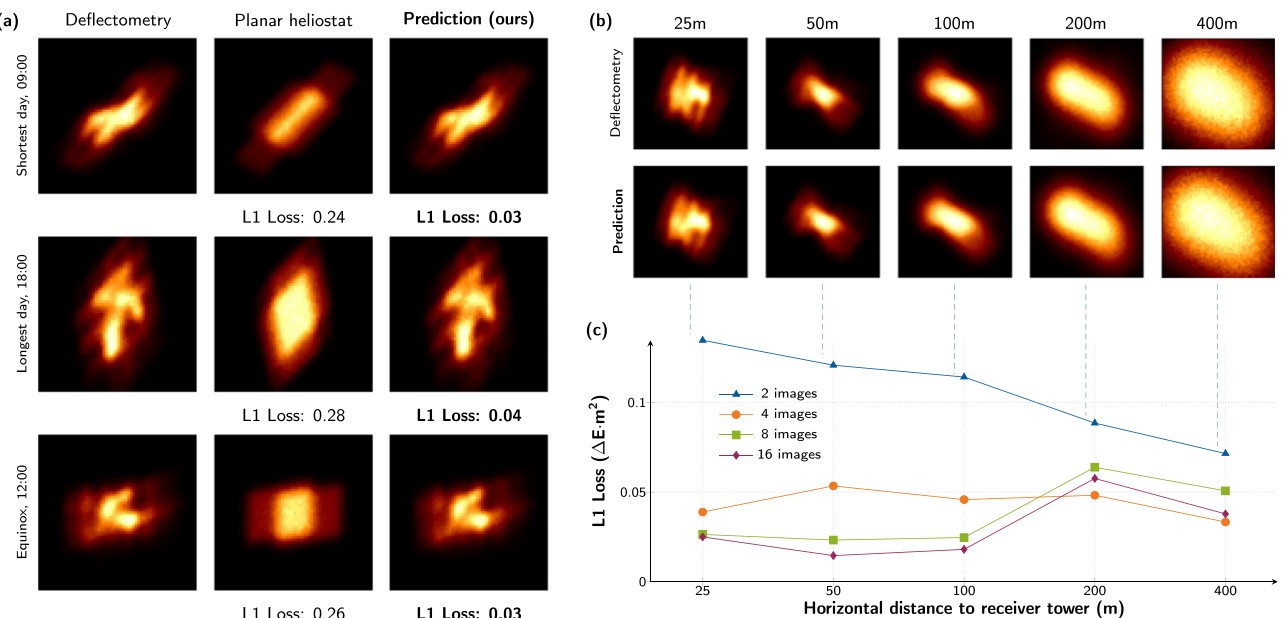

**Fig. 4 | Hyper-parameter study for the heliostat array. a** Comparison of the irradiance profile on three days of the year, i.e., the shortest, longest and at equinox. The ground truth is a simulation based on a deflectometry measured surface, contrasted with a perfectly planar heliostat and the proposed differentiable ray tracer trained on 16 images. Our method predicts complex structures accurately on a fine-grained level. **b** Simulated and inferred irradiance profiles at different distances between heliostat and calibration target. **c** Quantitative assessment of prediction quality for varying distances and number of training images. Comparisons in all subfigures are made with test images distinct from the training set. To facilitate an enhanced visual comparison of the focal spot details, the images were cropped. The depicted loss corresponds to the image of the complete calibration target. The numerical values can be taken from the Hyperparameter section in the Supplementary.

section in the Supplementary), we achieve a better irradiance prediction than the deflectometry procedure.

### Sensitivity analysis on deflectometry-initialized simulations

We have additionally studied the sensitivity of the proposed differentiable ray tracer's predictive performance with respect to variations of the non-learnable parameters, i.e., a heliostat's distance to the receiver, the position of the light source, as well as the number of images used in the learning process. The results are summarized in Fig. 4. For that purpose, we generate synthetic ground truths by simulating the heliostat geometry obtained from deflectometry measurements in various configurations that are all distinct from those used for training. The underlying assumption is that this measurement captures the true heliostat deformations to a sufficiently accurate degree. The heliostats in Jülich are astigmatically corrected and aligned with the target[78]. In this parameter study, we do not simulate this, since on the one hand it reduces the variance of the solution space and thus makes it easier to predict focal spots and on the other hand it is not implemented in all power plants. In subsequent simulations, we initialize virtual heliostats with the deflectometric deformations and compute the corresponding irradiance profiles while varying the non-trainable parameters.

The left panel (a) shows how the differentiable ray tracing approach performs on the shortest and the longest day of the year, as well as at equinox. These are the extremes of the sun's position over the course of the year and therefore pose particular challenges. Yet, when trained on 16 images, the prediction error of our differentiable ray tracer is roughly an order of magnitude lower compared to simulations with an idealized planar heliostat. Notably, our proposed approach is also able to nearly perfectly recreate the simulated deflectometry irradiance profiles.

In the upper right panel (b), the variation of distance to the receiver is depicted. The geometry obtained from deflectometry is used in simulations and compared against the prediction of the differentiable ray tracer trained on two images. As can be seen, the waviness of the focal spot decreases as the distance increases, resulting in blurred images. This phenomenon occurs because the focal spot is a convolution of the solar intensity distribution function and the mirror surface profile. At greater distances, the solar intensity function becomes increasingly dominant, resulting in a nearly circular focal spot as the distance approaches infinity. This also makes it more difficult to predict the surface. As will be shown in the following section, this is also the cause of failure for surface reconstruction at distances of over 200 m. Nevertheless, the differentiable ray tracer accurately captures the characteristic variations of the focal spots, largely independent of the distance. The differences in L1 loss are negligible and consistently less than $6 \times 10^{-2}$ using four or more images. The plot in panel (c) depicts the predictive performance while varying the heliostat distance to the target and the number of training images. Our method is able to reconstruct the focal spots with high precision with as few as *two training images*. A consistent leveled performance is achieved with four images. In all cases, the averaged test loss has been calculated on five held-out images not used during training. In summary, depending on the desired metrology resolution of the plant operator, roughly 4 images per heliostat would need to be captured. This would result in full-field resolution, with slight accuracy increases when using 8 images per heliostat.

### Heliostat surface reconstruction with learning NURBS

The high predictive performance of the differentiable ray tracer and its reliance on physical principles suggests that the learning NURBS surface model also results in a physically meaningful set of control points. Our experiments indicate that under favorable conditions, i.e., the heliostat being close to the receiver and reconstructed from multiple images, this holds and the real surface may be obtained (see Figs. 1a, 5d). However, due to the significant ill-posedness of the problem, e.g., overlap of the focal spots of the canted facets, or blurring with increasing receiver distance, it is highly likely that the supervised

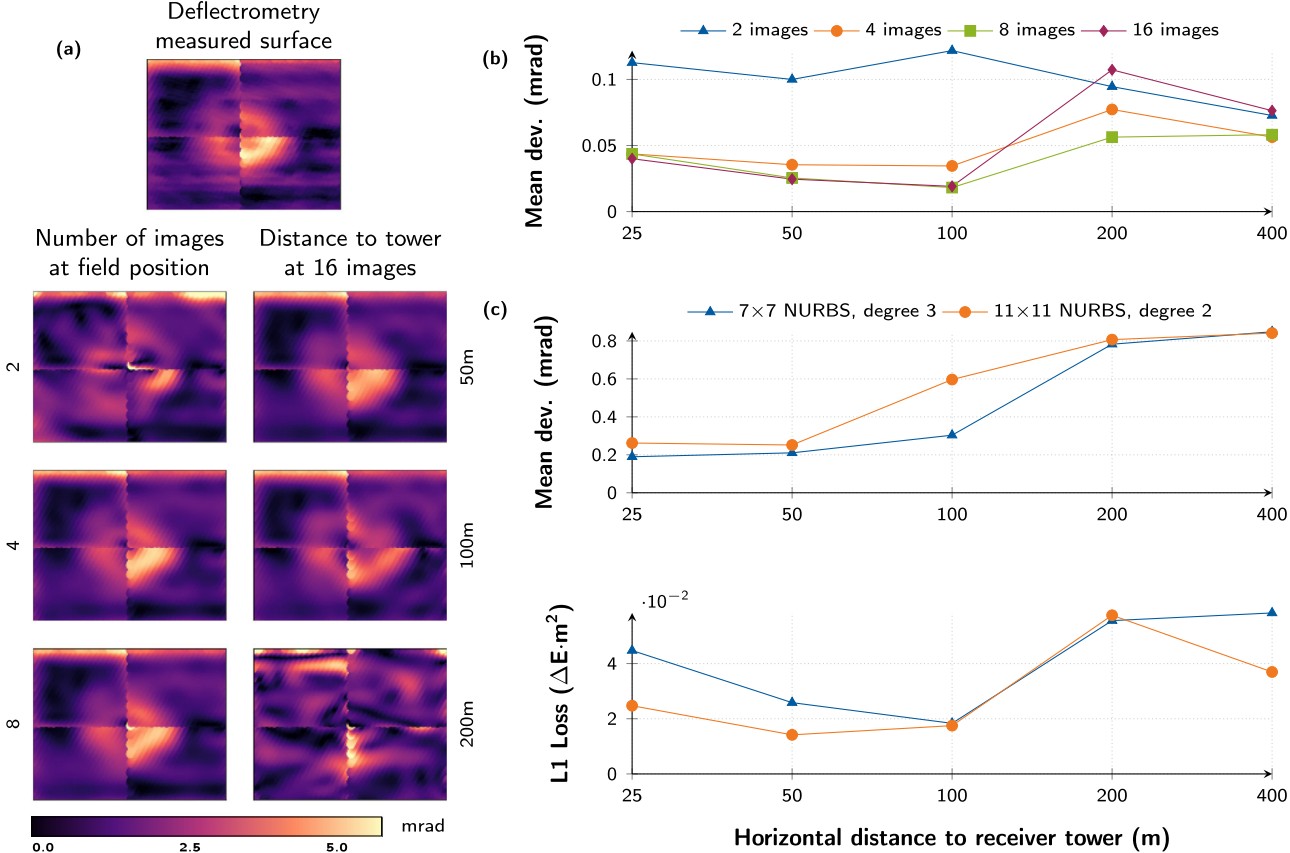

**Fig. 5 | Surface reconstruction capabilities and NURBS study. a** Heliostat surface reconstructions from focal spot images depending on their count (left column) and the distance to the receiver (right column). The imaged centered at the top shows the ground truth. The visualizations show deviations in mrad of the surface normals from the ideal planar surface. **b** Quantitative assessment of the reconstruction quality. For distances of up to 100 m, learning NURBS surfaces can be reconstructed accurately with four or more training images. **c** Surface reconstruction and irradiance quality with respect to the learning NURBS parameter count, polynomial degree, as well as the distance to the receiver. 11 × 11 learning NURBS excel at predicting irradiances, while 7 × 7 NURBS result in more precise surface reconstructions.

regression problem does not have a unique, physically meaningful minimum.

We once again consider the heliostat at 25 m north of the receiver and its deflectometry-measured deviation from the ideal planar surface. During the experimental preparation, we have purposely created a bump deformation by tightening the adjustment screws at the center of all four facets. This way, we can qualitatively judge surface reconstructions already at first glance. In doing so, we also ensured that the mean deviation is approximately 2 mm from a planar surface, which is in the range of typical heliostat surface defects.

The ground truth surface is depicted in Fig. 5a at the top. Below are two columns with reconstructions learned by our NURBS surface model. Reconstructions in the left column are obtained by varying the number of training data, while for the right column, the distance of the heliostat to the tower was altered. The results are summarized quantitatively in Fig. 5b. Learning NURBS can accurately infer the surface up to a heliostat-receiver distance of 100 m. Beyond that, the reconstruction quality quickly degrades. Interestingly, the precision of the irradiance reconstruction (see Fig. 4c) is barely affected. This is a consequence of the ill-posedness of the problem.

This metrology approach derives precise heliostat surface deformations from focal spots only. Even at heliostat-receiver distances of several dozen meters, the reconstructed surface model achieves sub-millimeter precision. Experiments with simpler hypothetical geometries reveal that the reconstructive performance is largely constrained by the canting ambiguity. Heliostats without canting or measurements taken outside of the focal plane raise the surface reconstruction ability

drastically. So far, we were not successful in bi-uniquely reconstructing surfaces directly from calibration imagery. We assume that we can alleviate this problem by (a) obtaining more data and (b) carefully tuning the learning process. Yet, we want to draw the attention to the fact that a precise surface reconstruction is not essential to the main objective of irradiance profile prediction.

The above experiments have additionally been repeated for different learning NURBS configurations. We find that for our heliostat type, 7 × 7 learning NURBS per facet with a cubic spline degree of $k = 3$ results in most optimal reconstructions. In contrast to that, 11 × 11 learning NURBS with a square spline degree of $k = 2$ are particularly well suited for flux density prediction. These results are summarized in Fig. 5c. These seemingly contradictory findings can be attributed to sweet spots in the number of trainable parameters. The smaller 7 × 7 NURBS configuration have a higher interlocking enabling the reconstruction of fine-grained surface features without significantly increasing the ill-posedness. However, the reduction in degrees of freedom compared to the 11 × 11 NURBS' higher number of parameters directly affects the predictive performance of focal spot images. A higher parameter count leads the optimization to better, but aphysical minima.

## Discussion

We have showcased the enormous potential of differentiable ray tracing for in-situ metrology and the optimization of CSPs. In the forward direction, our formulation provides a tangible improvement over the state-of-the-art in classical ray tracing in terms of required calculations.

In the backward direction, the differentiable ray tracer enables learning of physical characteristics of a modeled system. We demonstrated this by using images from the fully automated heliostat calibration to infer heliostat parameters that are otherwise difficult to measure. The derived information can then be utilized in classical ray-tracing simulations for automatic control purposes. The results from a field test at an actual CSP confirm that differentiable ray tracing yields high-quality irradiance predictions in practice. In-silico experiments indicate that the approach generalizes to entire heliostat fields with tens to hundreds of thousands of heliostats. Compared to deflectometry, we do not require difficult-to-meet measurement condition, simplifying operational procedures. The presented approach can be integrated into existing operating procedures with low to no additional cost through reuse of the current calibration infrastructure. Moreover, even on regular off-the-shelf computers, our approach is computationally fast enough to predict irradiance fluxes ahead of time. We have additionally sketched out our method's applicability beyond the specific use cases of surface reconstruction and irradiance prediction. Next to the discussed optimization of alignments for calibration, differentiable ray tracers at solar towers may also be used to improve canting, focus, shape or positioning of the heliostat, allowing for completely new applications, reaching from heliostat to field design. From a machine learning perspective, one of the major strengths of the proposed method is the combination of a physical simulation model with real-world data. On the one hand, the resulting model requires only a small number of trainable parameters, directly mapping to reality with sufficient detail, i.e., the learning NURBS. On the other hand, the amount of data necessary to train them is small due to the model being informed by the underlying physical principles. As a result, we can observe strong predictive performance of differentiable ray tracing even in ill-posed and underdetermined regimes. The approach can be easily extended to other use cases due to its flexibility. The physically motivated model combined with automatic differentiation techniques enables the easy integration of different geometries, e.g., heliostat archetypes, or diverse regularization techniques. Such adaptations permit the use of our method at a wide range of CSPs, possibly with different data quality and availability.

The currently employed physical model captures only a subset of the real-world effects. Some of the important, yet missing, factors are heliostat deformations, e.g., under gravitational load[79], or environmental influences like (partial) cloud[12] or fog cover, can affect irradiance prediction during data collection over the year. Additionally, soiling and air pollution may also negatively impact irradiance predictions and surface reconstruction. If such conditions were to be consistently measured, they could be accounted for. To mitigate these disturbances, it is necessary to expand the dataset to maintain predictive accuracy. The extent to which this affects both the dataset and the accuracy requires further investigation in future research. In our experimental setup, omitting these complexities is justified, due to clear sky weather conditions. In principle, these should be incorporated into the physical modeling pipeline whenever feasible. Practically, this is inherently difficult due to the high cost and complexity of the physical modeling approaches. For example, external conditions, such as wind, may not be entirely known or change over time. A possible alternative to this physically-informed design process is extended data-driven modeling. The differentiable ray tracing formulation facilitates the inclusion of other machine learning techniques. For example, a neural network could be used to predict the localized external conditions, e.g., wind gusts. Due to the differentiable ray tracer formulation, it is possible to train the physical and data-driven models simultaneously with gradient-based optimization. The degree to which both are leveraged can be tuned as part of the power plant engineering process.

This metrology bridges the gap between data-driven modeling and physical modeling in the field of CSPs. Thus, we firmly believe we have made a significant contribution in achieving one of the CSP metrology goals of the *Roadmap to Advance Heliostat Technologies for Concentrating Solar-Thermal Power Plants*[11]. Beyond CSPs, our proposed method may be an inspiration for other fields, such as the automotive or aircraft industry and non-line-of-sight imaging. Trainable, data-driven models that are not black boxes, but founded on physical principles can be robust, interpretable tools for improving real-world processes.

## Methods

### Differentiable ray tracing formalism

This section describes the key equations for our proposed differentiable ray tracer in depth. The main physical quantity is the radiance $L$, which describes the radiation field in terms of power per area and solid angle (W/ M²sr). It depends on the position $\vec{x}$ and the direction $\vec{t}$. In non-absorbing media, the radiance is constant along any line:

$$L\left(\vec{x}, \vec{t}\right) = L\left(\vec{x} + \lambda \vec{t}, \vec{t}\right) \qquad \forall \lambda. \tag{3}$$

parameterized by the scalar $\lambda$. The radiance field $L_\odot$ created by the sun and which is visible in direction $\vec{t}_\odot$ may be well approximated by a Gaussian distribution of the form

$$L_\odot \propto e^{-\left(\frac{\arccos \vec{t} \cdot \vec{t}_\odot}{\theta_\odot}\right)^2}, \tag{4}$$

with an aperture angle of $\theta_\odot = 0.00025°$. In order to obtain the irradiance $E\left(\vec{x}\right)$, the power per surface area at a surface position $\vec{x}$ is obtained by integration over the solid angle $\Omega$. This includes the cosine factor that is well known from, e.g., the rendering equation[80,81]:

$$E\left(\vec{x}\right) = \int_\Omega L\left(\vec{x}, \vec{t}_r\right) \vec{n}_c \cdot \vec{t} \quad d\Omega, \tag{5}$$

with $\vec{n}_c$ the normal vector of the calibration target and $\vec{t}$ the considered directional unit vector of the ray after the reflection at the heliostat. For a given point on the target, this integral can be evaluated in the following way. For each direction on the unit hemisphere, the corresponding intersection $\vec{h}$ with the heliostat is calculated. Based on the normal vector, the reflection towards the sun is calculated with the reflection matrix (see Equation (6)). Depending on the system's geometry, a large fraction of the evaluated direction vectors may either not intersect with the heliostat surface or will lead to directions with negligible solar radiance. We therefore transform the integral into a surface integral across the heliostat surface $A_h$,

$$E\left(\vec{x}\right) = 4\pi \int_A L_\odot\left(\vec{t}\right) \vec{n}_r \cdot \vec{t}_r \frac{\vec{n}_h \cdot \vec{t}_r}{\left\|\vec{x} - \vec{h}\right\|^2} \quad dA_h, \tag{6}$$

where $\vec{t}$ is the unit vector pointing from the target point $\vec{x}$ to the heliostat intersection point $\vec{h}$. We will call the directional vector, which is obtained by reflecting $\vec{t}$ on the surface, $\vec{t}_r$. On curved heliostats, the evaluation of $\vec{t}_r$ requires the heliostat normal $\vec{n}_h = [n_1, n_2, n_3]^t$ at position $\vec{h}$ and the construction of the reflection matrix $M(\vec{h})$ as follows:

$$M\left(\vec{h}\right) = \begin{pmatrix} 1 - 2n_1^2 & -2n_1 n_2 & -2n_1 n_3 \\ -2n_1 n_2 & 1 - 2n_2^2 & -2n_2 n_3 \\ -2n_1 n_3 & -2n_2 n_3 & 1 - 2n_3^2 \end{pmatrix}. \tag{7}$$

Then, the reflected direction is $\vec{t} = M \cdot \vec{t}_r$. By introducing the Dirac $\delta$-function, we can formalize the integration over all

directions $\vec{t}_r$:

$$E\left(\vec{x}\right) = c \int_{A_h} \int_{\Omega'} L_\odot\left(\vec{t}\right) \delta\left(\vec{x} - \vec{x}_t\right) d\Omega' dA_h, \tag{8}$$

where $\vec{x}_t$ is the intersection of an incident ray from the sun with the direction $\vec{t}_r$ on the target plane after reflection. Its evaluation is complex, as it not only depends on $\vec{t}$, but also on the heliostat point $\vec{h}$, the normal vector $\vec{n}$ and the position and orientation of the heliostat. Due to the large distance of the heliostat to the target and the small change of normal of the heliostat, the other terms in the integral can be considered constant and collapsed with the other prefactor into a coefficient $c$. The transformation leading to the double-integral seems at first like a mathematical trick, but has a simple physical interpretation. In the initial formulation, rays have been traced for all directions from the target surface to the sun. Now, for all positions $\vec{h}$ on the heliostat surface, we cast rays in all possible directions towards the sun $\vec{t}_r$ and evaluate each ray's contribution to the surface irradiance. The vast majority of directions $\vec{t}$ will still lead to a negligible irradiance contribution. Therefore, we evaluate the integral over incident direction through *importance sampling*[65]. All rays are sampled from a directional distribution proportional to $L_\odot$. In order to evaluate the surface integral, we discretize the heliostat with a rectangular grid. So far, the presented ray tracing scheme is well-known and commonly used in other state-of-the-art ray tracers[47–54]. The crucial step to achieve differentiability is the discretization of the target surface. For this, we interpret the $\delta$ function as a set of coefficients, which is non-zero for grid points in the vicinity of $\vec{x}_t$. We relax this hard binning scheme and propose a soft differentiable binning scheme instead. The idea illustrated in Fig. 2a. If $\vec{x}_{ij}$ is the grid point to the lower left of the intersection point $\vec{x}_t$, the ray is distributed to the four nearest neighbors:

$$\gamma_{i,j} = \left(1 - \Delta_x\right)\left(1 - \Delta_y\right)$$
$$\gamma_{i+1,j} = \Delta_x\left(1 - \Delta_y\right)$$
$$\gamma_{i,j+1} = \left(1 - \Delta_x\right)\Delta_y$$
$$\gamma_{i+1,j+1} = \Delta_x\Delta_y$$

where $\Delta_x$ and $\Delta_y$ measure the distance of $\vec{x}_t$ to $\vec{x}_{ij}$ in units of the grid constant. With this formulation, each ray carries an irradiance contribution that is differentiable with respect to the ray's direction. By introducing these coefficients, we conclude the final differentiable ray tracing formulation in Eq. (2).

## Surface model

In modeling the surfaces, we have aimed to replicate the conditions of the CSP in Jülich as close as possible to ease comparison with the field tests. Each heliostat has two angular degrees of freedom in its movement and consists out of four canted facets, i.e., nearly planar square surfaces arranged in a two by two grid. For the purpose of this work, we assume the canting angles to be fixed and known.

The heliostat's surface, or, more precisely, its local normal vector $\vec{n}_h$, is the decisive element for the direction in which rays are reflected. Due to the large distances between heliostat and target, the corresponding irradiance $E\left(\vec{x}\right)$ is highly sensitive to changes of $\vec{n}_h$. We are therefore modeling each facet initially as ideally planar. The placement of a facet in the heliostat's coordinate system is chosen such that given a solar position, the line connecting the sun and the heliostat is reflected exactly into the target.

Due to the physicality, and therefore, mechanical stiffness, of the reflective surface, it is justified to assume that curvature of the heliostat facets is marginal. Our smooth learning NURBS models therefore starts an optimization procedure with its control points set in a

singular plane. The NURBS surface is composed of different B-splines. Each point on the NURBS surface is thereby uniquely defined by the set of control points $P$, control point weights $W$, knot vectors $U$ and $V$ as follows[82]:

$$S = f(P, W, U, V). \tag{9}$$

A surface is parametrized by the variables $u$ and $v$, with $0 \le u, v \le 1$. For a given point $(u, v)$, the corresponding surface point in three-dimensional space is obtained as follows (also compare Fig. 2b):

$$S(u, v) = \frac{\sum_{i=0}^n \sum_{j=0}^m N_i^p(u) N_j^q(v) W_{ij} P_{ij}}{\sum_{i=0}^n \sum_{j=0}^m N_i^p(u) N_j^q(v) W_{ij}}. \tag{10}$$

In that, we assume a regular square grid of control points indexed by $i$ and $j$. The polynomials $N_i^p$ are defined recursively:

$$N_i^p(u) = \frac{u - u_i}{u_{i+p} - u_i} N_i^{p-1}(u)$$
$$+ \frac{u_{i+p+1} - u}{u_{i+p+1} - u_{i+1}} N_{i+1}^{p-1}(u) \tag{11}$$
$$N_i^0(u) = \begin{cases} 1 & \text{if } u_i \le u < u_{i+1} \\ 0 & \text{otherwise.} \end{cases}$$

$N_i$, $N_j$ are the B-spline basis functions in the representation of *Curry and Schoenberg*[83]. The degree of the polynomial can be chosen freely and determines how many nodes are affected by another node's variation. The smaller the degree, the more local the modification is. For a heliostat facet, a higher degree can therefore be interpreted as a regularizer counteracting strong local curvature. High learning NURBS degrees have proven useful in reconstructing surfaces from a small numbers of observed images. Tangential vectors of a learning NURBS surface can be obtained by computing the derivative with respect to $u$ and $v$ and corresponding normal vectors as their cross product.

Within the ray tracing environment, the initial NURBS surface is chosen so that the control points $P$ are evenly distributed over the heliostat's surface. This is schematically visualized in Fig. 2b by green points. For the ray tracing process, any number of points $M$ is sampled along the surface (blue points), subject to $P \ll M$. For several applications, we found it sufficient to set all weights to unity, effectively rendering our surface a conventional B-splines, and to keep the in-plane positions of the control points fixed.

## Loss formulation

Our loss function is defined in the following way:

$$L = \alpha_{\text{raw}} L_{\text{raw}} + \alpha_{\text{align}} L_{\text{align}} \tag{12}$$

where

$$L_{\text{raw}} = \frac{1}{n_i \cdot n_j} \sum_{i,j} \left| E_{ij} - \hat{E}_{ij} \right|^{p_{\text{raw}}}, \tag{13}$$

with $E_{ij}$ the irradiance of a ground truth image and $\hat{E}_{ij}$ the irradiance of an image obtained from ray tracing a learned heliostat, each at pixel position $(i, j)$ of an image of size $\left(n_i, n_j\right)$.

$$L_{\text{align}} = \frac{1}{|\vec{z}|} \sum_i |z_i - \hat{z}_i|^{p_{\text{align}}}, \tag{14}$$

with $\vec{z}$ and $\hat{\vec{z}}$ giving the ground truth and learned heliostat's alignment, respectively.

Focal spot images are normalized with regard to the target plane's area per pixel to keep the loss independent of the resolution of our image. For our predictions on real data, we used $\alpha_{\text{align}} = \alpha_{\text{raw}} = 1$ for

prealignment and $\alpha_{align} = 0$ and $\alpha_{raw} = 1$ for surface reconstruction. $p_{align} = p_{raw} = 1$ in both cases.

We additionally incorporated weight decay and other penalty terms, e.g., for missing the calibration target area. However, as these were not essential for achieving the results presented herein, their detailed description is omitted. These terms remain available in the Supplementary section Full Loss Formulation for potential application in other contexts.

## Data availability

Data for the results presented in this study is available at Zenodo (https://doi.org/10.5281/zenodo.11047453).

## Code availability

The code for results presented in this study is available at Zenodo (https://doi.org/10.5281/zenodo.11047453). Subsequent versions will be shared at https://github.com/ARTIST-Association/ARTIST.

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

## Acknowledgements
The authors would like to thank Marie Weiel and Kaleb Phipps for their valuable feedback in creating this manuscript and their continued contributions to the source code. This work is supported by the Helmholtz Association Initiative and Networking Fund through the Helmholtz AI platform, and the HAICORE@JSC grant.

## Author contributions
M.P., J.E., S.K. conceived the study. M.P., J.E., S.K. developed the methodology. M.P., J.E. implemented the corresponding software. M.P., J.E., S.K. conducted the field experiments. All authors discussed the results. M.G., M.P., J.E., S.K. wrote the manuscript and designed the figures. D.MQ., R.PP. supervised and organized the funding. All authors read and approved the manuscript.

## Funding

## Competing interests
The authors declare no competing interests.
