## [Peer Review File · Nature Communications]

Automatic heliostat learning for in-situ concentrating solar power plant metrology with differentiable ray tracingREVIEWER COMMENTS

Reviewer #1 (Remarks to the Author):

The authors presented a methodology to reconstruct and estimate heliostat focus spot flux distribution using calibration images and machine learning.

While the application of differentiable ray tracing to heliostats is new, the idea of using calibration images to predict heliostat focus spot flux distribution is not new.

Also, the title of the manuscript is very misleading. It mentioned CSP optimization, while no optimization problem is studied other than optimizing the model with respect to data. Typical CSP optimization problems are CSP plant, heliostat field optimal design, and aiming strategy optimization, etc. It is recommended that the authors should modify the title to reflect the actual content of the work.

And because the authors only investigated the fitting of the data, important aspects of how to leverage heliostat surface models are not properly considered. Such as the amount of data that need to be collected for a decent size field, as well as deformation of the surface under different pose and wind conditions, and error induced by imaging etc.

Therefore, the overall novelty and significance of the original contribution of this work is substantially below other nature communications articles published. It is recommended that the authors conduct a thorough literature study of related work, modify the title, add comparative study with other existing differentiable ray tracing methods and submit to other journals.

Reviewer #2 (Remarks to the Author):

This paper deals with a challenging problem in thermosolar plants with central receiver systems. The authors develop a differentiable ray tracing machine learning approach that can derive the irradiance distribution of heliostats in a data-driven manner from a small number of calibration images. The approach is interesting and the paper is well written. The inclusion of differentiable ray tracer and the obtaining of a differentiable heliostat surface model seems to be coherent and the contribution is highly relevant.

Main comments:

How the calibration shown in figure 1 is affected by environmental conditions (irradiance, presence of clouds or fog, air contamination, mirrors reflectivity, etc.)? What is shown in the paper is the calibration and validation of heliostats very close to the target and simulations up to 400 m. Is the method applicable to heliostats located far away (even km) from the target?

The comparison of the developed ray tracing algorithm is done against STRAL. There are other ray tracking software that are commonly used in this kind of facilities, such as SolTrace, HFLD, Tonatiuh, GPU Ray Tracer. The suitability of STRAL for comparison purposes (instead of other packages) has been commented, but, it is really the best option for comparison purposes in this case?

The shape of the reflected radiation changes during operation and depending on the relative Sun-Earth position. Are all training images in Figure 4 obtained at the same data and time instants? The authors state that their method is able to reconstruct the focal spots with high precision with as few as two training images, but (I have probably missed something), can the shape be inferred at any moment to be used for automatic control purposes? Something is commented about in the Heliostat Surface Reconstruction section, but a deeper explanation should be welcome.

The approach uses a kinematic model of the heliostat which can be subject to modeling errors and uncertainty. How is this compensated by the algorithm?

The authors claim that to the best of their knowledge, we are the first to propose a metrology approach that is able to derive heliostat surface deformations from focal spots only. What are the advantages when compared to the flux mapping approach by, for instance, the paper from IMDEA Energy <https://doi.org/10.1016/j.solener.2023.112162>. Besides this work, the authors state that the approach serves to “predict heliostat-specific irradiance profiles, exceeding the precision of the state-of-the-art and establishing full automation”. There are other approaches aimed at closing the loop using cameras and deep learning approaches allowing online determination of the presence of clouds and closed-loop control of the entire solar field (works at Plataforma Solar de Almeria by Jose A. Carballo and coworkers or the Scalable heliostat calibration system (SHORT) developed at CENER) or by using synthetic models obtained from ray-tracing techniques and images taken in the real field and implemented using artificial intelligence methods for control purposes (works by Nicolas C. Cruz and coworkers from University of Almeria or Eduardo F. Camacho and coworkers from University of Seville) that are not considered in the statements made by the authors. Another examples are the works by Dong Li and coworkers en Zhejiang University in China or those by the Heliostat Consortium (HelioCon).

Therefore, I miss a critical and comparative analysis of the advantages and disadvantages of the authors’ approach to those recently published in the scientific literature.

Uncomplete references: The title of the articles is missing in most references.

Reviewer #3 (Remarks to the Author):

The author/s presented a differential ray tracing machine-learning approach for optimizing irradiance distribution and correct aiming strategy. The method is novel and has an enormous scope in this area. The manuscript is well-written, and an in-depth study has been carried out. The results are promising and support the concluding statements. My observations and reviews are as follows to improve the quality of the manuscript.

1. The abstract needs to be revised and made more technical in terms of the methodology and outcomes of the study. A brief statement of achieved results must be included in the abstract.
2. The literature review needs to be more comprehensive. More references for aim point strategies, camera target method, deflectometry, Monte Carlo ray tracing, and other optical testing methods would help understand the baselines for the current study.
3. The quality of Figures 1 (d) and (e) can be improved, or these two figures can be plotted separately with different Figure names.
4. The tracking mode significantly influences the image generation on the receiver plane. State the tracking method (Azimuth elevation, spinning elevation) used in this study and its effect on the reflected image.
5. Also, literature support for the tracking methods-based aiming strategy would help understand the impact on flux distribution or image generation.
6. Mention the optical properties used in this research.

Notes on revision made to manuscript NCOMMS-23-47231A-Z

The authors would like to thank the reviewers for their constructive comments and suggestions. We have used them to improve the quality of this manuscript. The manuscript has undergone a thorough revision according to the editor's and reviewers' comments and we feel it has improved in terms of width and depth. Furthermore, we have integrated additional related work with the overall aim of covering the body of study in more detail. For more details, please see below for our individual responses. For the reviewers' convenience, we have highlighted our edits and replies in blue.

Response to the Reviewers

Reviewer 1

Reviewer Comment 1.1 — The authors presented a methodology to reconstruct and estimate heliostat focus spot flux distribution using calibration images and machine learning. While the application of differentiable ray tracing to heliostats is new, the idea of using calibration images to predict heliostat focus spot flux distribution is not new.

Reply: Thank you for your comments on our manuscript regarding the methodology we presented for reconstructing and estimating heliostat focus spot flux distribution using calibration images and machine learning. We appreciate your acknowledgment of the novelty of applying differentiable ray tracing to heliostats. To address your statement about the use of calibration images to predict heliostat focus spot flux distribution not being new, we added similar approaches to our state of the art section. However, we demonstrate a big step ahead regarding practical application and accuracy of these approaches. Moreover, we sketch that our method is capable of a broad range of tasks other than surface reconstruction, such as heliostat calibration. The most closely related publication, suggested by the reviewers, is from Martínez-Hernández et al. (December 2023). We were not aware of this work, due to it being published after our submission to this journal. However, we would like to stress that we released a preprint of our work 9 months prior to Martínez-Hernández et al. For a more detailed discussion on this paper, we refer to our response 2.7.

We value your feedback and look forward to further dialogue on this matter.

Reviewer Comment 1.2 — Also, the title of the manuscript is very misleading. It mentioned CSP optimization, while no optimization problem is studied other than optimizing the model with respect to data. Typical CSP optimization problems are CSP plant, heliostat field optimal design, and aiming strategy optimization, etc. It is recommended that the authors should modify the title to reflect the actual content of the work.

Reply: We acknowledge that optimization is an ambiguous term in this context and may be interpreted as the optimization of the CSP yields. Hence, we propose to modify the title from "Optimization" to "Metrology" instead.

Reviewer Comment 1.3 — And because the authors only investigated the fitting of the data, important aspects of how to leverage heliostat surface models are not properly considered. Such as the amount of data that need to be collected for a decent size field, as well as deformation of the surface under different pose and wind conditions, and error induced by imagining etc.

Reply: We apologize that this point did not come across clearly. Our study shows that only 4 images per heliostat are required to obtain acceptable flux predictions. With an increasing number of images, the prediction quality can be improved. If a full-field resolution is desired, one would thus require $\#heliostats \times 4$ images in total. This is in a range of number of calibration images which are already measured during build-up of the plant to initially align the heliostats. For this statement, we reference the quantitative evaluations in Figure 4 (c) and the final paragraph of the Section “Sensitivity Analysis on Deflectometry-Initialized Simulations”. For clarity, we added a short summary about the amount of data needed for the heliostat field.

Reviewer Comment 1.4 — It is recommended that the authors conduct a thorough literature study of related work, modify the title, add comparative study with other existing differentiable ray tracing methods [...].

Reply: We have expanded the body of related work. Please refer to our reply to comment 8 of reviewer #2.

Reviewer Comment 1.5 — Therefore, the overall novelty and significance of the original contribution of this work is substantially below other nature communications articles published. It is recommended that the authors conduct a thorough literature study of related work, modify the title, add comparative study with other existing differentiable ray tracing methods and submit to other journals.

Reply: We accept criticism regarding technical aspects of the manuscript and thanks to the reviews have substantially improved it. But we would like to emphasize the significance and novelty of our work again: in our paper we publish a novel method to derive realistic surfaces up to 100m away from the calibration target with sub-millimeter accuracy. This requires just 4 focal spot images from an existing fully automated routine at the solar tower. Every obtained surface can be used for high quality year-round focal spot prediction for every heliostat in the solar field. The conducted experiments exceed the accuracy of the irradiance prediction by deflectometry for the whole year. Thus, we provide an alternative to deflectometry, a high-precision measurement technique commonly referenced in the scientific literature, but which has been hindered from practical implementation in power plant operations over decades due to its inherent unreliability. Instead, we use an existing measurement, widely available in all power plants, to exceed the irradiance prediction accuracy of this scientific state-of-the-art technique. Moreover, in upcoming works the method can be used to determine the exact orientation of the heliostats and other relevant properties (position, canting, etc.). The most similar publication known to us is limited to surface prediction only, has a maximum target-heliostat distance of 10m and was published 9 months after the first version of our preprint. Regarding the use of differentiable ray tracing methods, we are the first to apply this approach in the context of CSPs and clearly show its remarkable potential. However, we use only elementary concepts from *differentiable ray tracing*, but our method is an extension of methods typical for CSP/heliostat simulation, e.g. the use of importance sampling), modified for differentiability. We hope that the manuscript improvements and a reconsideration of our contribution can lead to a more positive assessment and would like to thank reviewer #1 again for their efforts.

Reviewer 2

Comments to the Author:

This paper deals with a challenging problem in thermosolar plants with central receiver systems. The authors develop a differentiable ray tracing machine learning approach that can derive the irradiance distribution of heliostats in a data-driven manner from a small number of calibration images. The approach is interesting and the paper is well written. The inclusion of differentiable ray tracer and the obtaining of a differentiable heliostat surface model seems to be coherent and the contribution is highly relevant.

Reviewer Comment 2.1 — How the calibration shown in figure 1 is affected by environmental conditions (irradiance, presence of clouds or fog, air contamination, mirrors reflectivity, etc.)?

Reply: Thank you for your inquiry. We acknowledge omitting the discussion on how weather conditions influence our method. For the first prototype of the differentiable ray tracer, we ensured optimal weather conditions to showcase the method's principle viability. We recognize the importance of further studying the impact of external environmental conditions. In fact, we are currently investigating such effects in a separate study, which has unfortunately not yet reached sufficient maturity.

Regarding surface reconstruction under partial shading from clouds or fog, our algorithm seeks to optimally represent the imagery from our training set. Such shading or distortions in part of the imagery could lead to deviations in predictions throughout the year. However, a larger dataset, e.g., comprising 16 images or more, could minimize this effect and effectively result in an "average" surface under varying conditions.

As for air pollution or heliostat soiling, air pollution, uniformly reducing the air's refractive index should negligibly affect irradiance prediction but may limit a heliostat's maximum effective distance to the tower for accurate surface reconstruction. Heliostat soiling warrants further thought. Similar to air pollution, uniform pollution has a minimal impact and can be easily added. Extensive non-uniform contamination akin to cloud shading, e.g., spills, requires considerably more effort in the ray tracer modeling.

We have amended the main document which incorporates the above analysis as a paragraph in the "Discussion" section.

Reviewer Comment 2.2 — What is shown in the paper is the calibration and validation of heliostats very close to the target and simulations up to 400 m. Is the method applicable to heliostats located far away (even km) from the target?

Reply: The irradiance image is a convolution of the solar function and the heliostat surface function, with respect to the the distance between the heliostat and the tower. For the applicability of our approach, it is necessary to compare the characteristic length ratios, or corresponding angles. Three ratios are relevant in so far: a) the ratio of heliostat size to heliostat distance from the tower or the angular size of the heliostat when viewed from the tower; b) the magnitude of heliostat deformation compared to the size, or equivalently the angular deviation from an ideal plane; c) the ratio of distance to the sun and its distance, equivalently the angular size of the solar disc. For correspondingly larger heliostats at km-distances with deformations of comparable magnitude, our method would be equally applicable. Since you raised this question, we added the predictions for 800 m-1600 m meters in our supplementary.

Reviewer Comment 2.3 — The comparison of the developed ray tracing algorithm is done against STRAL. There are other ray tracing software that are commonly used in this kind of facilities, such as SolTrace, HFLD, Tonatiuh, GPU Ray Tracer. The suitability of STRAL for comparison purposes (instead of other packages) has been commented, but, it is really the best option for comparison purposes in this case?

Reply: We chose STRAL for our study for two main reasons. First, the accuracy of STRAL has been proven in several prior works [1, 2]. Upon its initial release, it was for example tested with deflectometry data from the CAESAR 1 power plant. Research by Belhomme et al. [3] showed that STRAL can simulate solar focal spots with nearly 98% overlap between simulation and measurement [4]. Osorio et al. [5] wrote: “The main purpose of [STRAL] is the flux density simulation of heliostat fields with a very high accuracy in a small amount of computation time. The software is primarily designed to process real sun shape distributions and real highly resolved heliostat geometry data”. This capability is consistent with the goal of our study to accurately compare deflectometric simulations with real-world scenarios, making STRAL an optimal choice. Since STRAL is used for fast processing of certain types of data as described above. Our ray tracer shows a slight computational processing advantage in this specific use case, which prompted us to show a runtime comparison to STRAL.

Thank you for pointing this out, we added our reasoning in Section “Comparison with Classical Ray Tracing”

Reviewer Comment 2.4 — The shape of the reflected radiation changes during operation and depending on the relative Sun-Earth position. Are all training images in Figure 4 obtained at the same data and time instants?

Reply: Thank you for your feedback and for pointing out the confusion regarding the images presented in Fig. 4. We apologize for any misunderstandings. To clarify, the images depicted in Fig. 4a are indeed test images, not training images. Our ray tracer was optimized using a distinct set of 16 images, each selected from random timestamps. The specific images showcased in Fig. 4a were chosen to illustrate the ray tracer’s reconstruction capabilities across different periods of the year. These images represent the shortest day, the equinox, and the longest day, taken at three different times of the day: morning, noon, and evening. This selection was made to demonstrate the robustness of our approach in handling diverse lighting and positional challenges posed by the Sun’s movement throughout the year. Furthermore, Fig. 4b focuses on the reconstruction capabilities for a single heliostat at a specific date and time in different distances to the tower. The shown image was reconstructed at a randomly selected date/time and is distinct from the training dataset. Lastly, Fig. 4c compares the reconstruction capability with respect to the number of training images used. The shown loss refers to the reconstruction capabilities for a distinct test image.

We added information about depicted images being test data to manuscript and caption

Reviewer Comment 2.5 — The authors state that their method is able to reconstruct the focal spots with high precision with as few as two training images, but (I have probably missed something), can the shape be inferred at any moment to be used for automatic control purposes? Something is commented about in the Heliostat Surface Reconstruction section, but a deeper explanation should be welcome.

Reply: We hope that our above response has already clarified the origin of this question, but to address your query more comprehensively: yes, our method indeed enables the generation of reasonably accurate

irradiance profiles throughout the entire year using just two training images. Once the surface shape has been obtained by using our approach – this process takes about 15 minutes on a consumer-grade GPU – this information can be utilized in classical ray tracing simulations for automatic control purposes. We hope this clarifies your question.

We added this to the *Discussion* section.

Reviewer Comment 2.6 — The approach uses a kinematic model of the heliostat which can be subject to modeling errors and uncertainty. How is this compensated by the algorithm?

Reply: The heliostats do indeed deviate from the specified target point due to their tracking error. As a result, an ideally aligned heliostat would hit the target at a different position than a real heliostat. E.g. during heliostat calibration, a geometry model is created, which contains error parameters that can describe this deviation. The parameters are mathematically adjusted to minimize the deviation of the target points. However, a residual error usually remains – about 2 mrad. To reduce this error, we do not use a geometry model that is valid for all sun angles. Instead, we perform a pre-training step in which the orientation of the heliostat is optimized individually for each data point (training and test). We initialize an ideal heliostat aligned with the center of the target and iteratively adjust its orientation on a single image using our optimization pipeline. This is done until the ideal focal point and the real focal point overlap as closely as possible. The vectors used to align the heliostat are determined for each individual image and stored for later surface training. This ensures that the model error in the alignment is as small as possible.

It is important to note that the measured and ideal focal points are still not perfectly congruent. The accuracy is albeit sufficient for the NURBS-based learning process to compensate for the remaining discrepancies. While not used in this work, a realistic geometry model may also be used for this purpose, as is common in heliostat calibration. This will allow the surface error and the alignment error to be determined in the same optimization routine and improve each other. We intend to study these models in future work.

Reviewer Comment 2.7 — The authors claim that to the best of their knowledge, we are the first to propose a metrology approach that is able to derive heliostat surface deformations from focal spots only. What are the advantages when compared to the flux mapping approach by, for instance, the paper from IMDEA Energy <https://doi.org/10.1016/j.solener.2023.112162>.

Reply: We thank the reviewer for making us aware of this manuscript by Martínez-Hernández et al.[6]. The paper is indeed highly related to our manuscript. We were not aware of this work, due to it being published after our submission to Nature Communications. It was therefore impossible to refer to.

Martínez-Hernández's publication describes a similar measurement setup to reconstruct heliostat surfaces from focal spots. Unlike differentiable ray tracing and a continuous NURBS surface, however, their work uses a numerical algorithm to determine the Pearson coefficient and a heliostat described by a mesh made from discrete surface elements. They are able to reconstruct surfaces in an optimal distance of 5 m up to a reasonable accuracy within 10 m. For practical use cases, this distance is too low for actual surface metrology in real setups, thus, a measurement setup is used which inhabits a moving target.

The short reconstruction distances are most likely due to the missing intersection between the surface elements. In our method this is surpassed by the NURBS surface where the surface points are interlocked with each other and form a continuous surface. Thus, our methods integrates directly into the existing heliostat calibration procedure and enables stable surface reconstructions at limits of up to 100 m, without

any moving target (in Jülich, this covers the first half of the field). Moreover, the irradiance prediction is nearly independent of a correct surface reconstruction as shown in Fig. 4 and 5. The obtained surfaces in distances higher than 100 m can not be used for surface diagnosis, but they can still be used for ray tracing to obtain more realistic irradiance profiles, which is far more important information for the power plant operation.

Moreover, we still believe that our claim to be the very first to derive heliostat surface deformations from focal spots is still valid. While the publication by Martínez-Hernández appeared earlier in a journal than ours, we published the results presented in this publication as a preprint already in February 2023. We will be happy to send you the corresponding link, but it is important to us to comply with the blinding procedure. We are open to the editor's decision on how to proceed in this regard, whether by sharing the preprint link with you or by any other approach the editor deems appropriate.

Lastly, our method is capable of also inferring e.g heliostat alignment, optimal aimpoint distribution or canting error with just switching the optimizable parameter and thus can be used for a broad range of applications.

We will happily incorporate this work into our related work section to provide a comprehensive overview of advancements in the field, since the topicality of this release shows the urgency of methods for surface reconstruction and irradiance prediction.

Reviewer Comment 2.8 — Besides this work, the authors state that the approach serves to “predict heliostat-specific irradiance profiles, exceeding the precision of the state-of-the-art and establishing full automation”. There are other approaches aimed at closing the loop using cameras and deep learning approaches allowing online determination of the presence of clouds and closed-loop control of the entire solar field (works at Plataforma Solar de Almeria by Jose A. Carballo and coworkers or the Scalable heliostat calibration system (SHORT) developed at CENER) or by using synthetic models obtained from ray-tracing techniques and images taken in the real field and implemented using artificial intelligence methods for control purposes (works by Nicolas C. Cruz and coworkers from University of Almeria or Eduardo F. Camacho and coworkers from University of Seville) that are not considered in the statements made by the authors. Another examples are the works by Dong Li and coworkers en Zhejiang University in China or those by the Heliostat Consortium (HelioCon). Therefore, I miss a critical and comparative analysis of the advantages and disadvantages of the authors' approach to those recently published in the scientific literature.

Reply: Based on your comments, we have identified the following publications that are most in line with your description. We would like to briefly comment on them here:

1. **heliostat surface optimization by irradiance profiles and aim point strategies:** Thank you for your hints, analyzing this works we clearly see the progress in this development. We added all following publications to a new paragraph in our introduction.
 - (a) **Determination of heliostat canting errors via deterministic optimization[7]:** Thank you for this hint, this is maybe the first approach to optimize heliostat parameters according to heliostat focal spots.
 - (b) **Heliostat field aiming strategy optimization with post-installation calibration[8]:** A few years later, beside the canting it is now possible to also adapt the heliostat surface by 4 parameters per facet.

- (c) **Advanced surface reconstruction method for solar reflective concentrators by flux mapping[6]:** They optimized an heliostat surface model with hundreds of parameters achieving deflectometric-like accuracy. However, the maximum distance for reconstruction is greatly reduced.
 - (d) **Real-time heliostat field aiming strategy generation for varying cloud shadowing using deep learning[9]: Finding an optimal Aimpoint distribution for heliostats including cloud covering using deep learning.**
2. **Machine Learning supported heliostat calibration techniques:** While not directly correlated to our publication the following works show the advantage of new machine and deep learning techniques to the area of solar tower research. We mentioned them briefly in the introduction section.
- (a) **Validation of a low-cost camera for Scalable HeliOstat calibRation sysTem (SHORT)[10]:** An advanced Heliostat Calibration technique showing good tracking accuracy using cameras attached to each heliostat. However, the method is substantially different to the calibration technique used today in most commercial power plants.
 - (b) **New approach for solar tracking systems based on computer vision, low cost hardware and deep learning[11]:** also an interesting technique for heliostat alignment using cameras on each heliostat. Computer vision is used to detect the position of the sun and the focal spot on the calibration target.
 - (c) **A two-layered solution for automatic heliostat aiming[12]:** In the first layer a genetic algorithm is used to choose which heliostats and aimpoints are optimally used. The second layer is used to aimpoint optimization using gradient descent to achieve specific distribution. As shown in by [8], this method can profit from more accurate irradiance profile. So we think our approach is very helpful here.
3. **Model predictive control using machine learning approaches:** As the group before, the following publications show the advantage of data driven algorithms for the application at solar towers. The correlation to our publication is small but still noticeable.
- (a) **Model predictive control based on deep learning for solar parabolic trough plants [13]:** an MLP is used to avoid an MPC Controller. The NN shows some benefits. The biggest advantage seems to be the calculation time.
 - (b) **A fast implementation of coalitional model predictive controllers based on machine learning: Application to solar power plants [14]:** Again another good example for the advantages of Machine Learning in the field of solar power plants.
 - (c) **Structured Light Based High Precision 3D Measurement and Workpiece Pose Estimation [15]:** An advanced deflectometric method including pose estimation. Most likely more stable than classical deflectometric measurements.

Thank you for indicating these publications. If we have missed any paper you were referring to, please let us know and we will consider them.

Reviewer Comment 2.9 — Uncomplete references: The title of the articles is missing in most references.

Reply: We thank the reviewer for pointing out this mistake. Through a change of the LaTeX code the titles should now be displayed correctly.

Reviewer 3

Comments to the Author:

The author/s presented a differential ray tracing machine-learning approach for optimizing irradiance distribution and correct aiming strategy. The method is novel and has an enormous scope in this area. The manuscript is well-written, and an in-depth study has been carried out. The results are promising and support the concluding statements. My observations and reviews are as follows to improve the quality of the manuscript.

Reviewer Comment 3.1 — The abstract needs to be revised and made more technical in terms of the methodology and outcomes of the study. A brief statement of achieved results must be included in the abstract.

Reply: The abstract in its current form is written for the widest possible audience, as is usual for Nature Communications. We adhere to the standards set by Nature Communications:

“The abstract - [...] should serve both as a general introduction to the topic and as a brief, non-technical summary of the main findings and their implications.”

We do not give specific results here, but already refer to the successes achieved: “determine sub-millimeter imperfections”, “predict heliostat-specific irradiance profiles, exceeding the precision of the state-of-the-art”

We believe that maintaining the current abstract is appropriate.

Reviewer Comment 3.2 — The literature review needs to be more comprehensive. More references for aim point strategies, camera target method, deflectometry, Monte Carlo ray tracing, and other optical testing methods would help understand the baselines for the current study. [...] Also, literature support for the tracking methods-based aiming strategy would help understand the impact on flux distribution or image generation.

Reply: Thank you, we limited our submission to discussing methods directly related to our method. We fully agree that a broader classification helps understanding, so we have extended our introduction and ray tracing part with relevant sources and additional analysis.

Reviewer Comment 3.3 — The quality of Figures 1 (d) and (e) can be improved, or these two figures can be plotted separately with different Figure names.

Reply: According to your recommendations, we have made several enhancements to improve the quality and clarity of our figures:

We have incorporated the target image into Figure 1c to provide better context and aid in interpretation. In response to your suggestion, also enlarged the labeling in Figure 1d and to address concerns about Figure 1e, we have relocated it to Figure 3b. This separation not only allows for clearer visualization but also provides a more logical grouping of related information.

Reviewer Comment 3.4 — The tracking mode significantly influences the image generation on the receiver plane. State the tracking method (Azimuth elevation, spinning elevation) used in this study and its effect on the reflected image.

Reply: Thank you very much for pointing this out, we missed to describe a very crucial point here. The heliostats in Jülich use a primary horizontally axis and a perpendicular secondary axis. The mirrors of are astigmatically corrected and aligned with the target [16]. The simulations in our paper are limited to surface reconstruction and do not use a geometry model for alignment that would have to account for this heliostat structure. Nevertheless, the target-aligned astigmatism has an impact on the results. In our real experiments on the solar tower, the astigmatism correction leads to a reduced solution space (the target images are more similar to each other at different sun positions). In our in-silico simulations, we have omitted this astigmatism to make the prediction of the focal points for the ray tracer more difficult and to demonstrate its generalizability to other heliostats. We have added a reference to this in the text.

Reviewer Comment 3.5 — Mention the optical properties used in this research.

Reply: The reviewer has raised an important point about a better understanding of the environmental conditions of the experiments. Hence, we have added a new section with the optical properties that we deemed important to the supplementary material.

References

- [1] N. Ahlbrink, B. Belhomme, R. Flesch, D. Maldonado Quinto, A. Rong, and P. Schwarzbözl, “Stral: Fast ray tracing software with tool coupling capabilities for high-precision simulations of solar thermal power plants,” in *Proceedings of the SolarPACES 2012 conference*, 2012.
- [2] N. Ahlbrink, B. Belhomme, and R. Pitz-Paal, “Modeling and simulation of a solar tower power plant with open volumetric air receiver,” in *Proceedings 7th Modelica conference*, 2009, pp. 20–22. DOI: 10.3384/ecp09430048.
- [3] B. Belhomme, R. Pitz-Paal, P. Schwarzbözl, and S. Ulmer, “A new fast ray tracing tool for high-precision simulation of heliostat fields,” *Journal of Solar Energy Engineering*, vol. 131, no. 3, 2009. DOI: 10.1115/1.3139139.
- [4] B. Belhomme, *Bewertung und Optimierung von Zielpunktstrategien für solare Turmkraftwerke*. Shaker, 2011, ISBN: 978-3844002591.
- [5] T. Osório, P. Horta, M. Larcher, R. Pujol-Nadal, J. Hertel, D. W. Van Rooyen, A. Heimsath, S. Schneider, D. Benitez, A. Frein, *et al.*, “Ray-tracing software comparison for linear focusing solar collectors,” in *AIP conference proceedings*, AIP Publishing, vol. 1734, 2016. DOI: 10.1063/1.4949041.
- [6] A. Martínez-Hernández, R. Conceição, C.-A. Asselineau, M. Romero, and J. González-Aguilar, “Advanced surface reconstruction method for solar reflective concentrators by flux mapping,” *Solar Energy*, vol. 266, p. 112 162, 2023, ISSN: 0038-092X. DOI: 10.1016/j.solener.2023.112162.

- [7] A. Sánchez-González, C. Caliot, A. Ferrière, and D. Santana, “Determination of heliostat canting errors via deterministic optimization,” *Solar Energy*, vol. 150, pp. 136–146, 2017. DOI: 10.1016/j.solener.2017.04.039.
- [8] R. Zhu, D. Ni, T. Yang, J. Yang, J. Chen, and G. Xiao, “Heliostat field aiming strategy optimization with post-installation calibration,” *Applied Thermal Engineering*, vol. 202, p. 117720, 2022. DOI: 10.1016/j.applthermaleng.2021.117720.
- [9] S. Wu and D. Ni, “Real-time heliostat field aiming strategy generation for varying cloud shadowing using deep learning,” in *AIP Conference Proceedings*, AIP Publishing, vol. 2815, 2023. DOI: 10.1063/5.0149189.
- [10] I. Les, A. Peña-Lapuente, M. Sanchez, D. Olasolo, C. Villasante, R. Enrique, and J. Fernandez-Reche, “Validation of a low-cost camera for scalable heliostat calibration system (short),” in *AIP Conference Proceedings*, AIP Publishing, vol. 2445, 2022. DOI: 10.1063/5.0085764.
- [11] J. A. Carballo, J. Bonilla, M. Berenguel, J. Fernández-Reche, and G. Garcia, “New approach for solar tracking systems based on computer vision, low cost hardware and deep learning,” *Renewable energy*, vol. 133, pp. 1158–1166, 2019. DOI: 10.1016/j.renene.2018.08.101.
- [12] N. C. Cruz, J. D. Álvarez, J. L. Redondo, M. Berenguel, and P. M. Ortigosa, “A two-layered solution for automatic heliostat aiming,” *Engineering Applications of Artificial Intelligence*, vol. 72, pp. 253–266, 2018. DOI: 10.1016/j.engappai.2018.04.014.
- [13] S. Ruiz-Moreno, J. R. D. Frejo, and E. F. Camacho, “Model predictive control based on deep learning for solar parabolic-trough plants,” *Renewable Energy*, vol. 180, pp. 193–202, 2021. DOI: 10.1016/j.renene.2021.08.058.
- [14] E. Masero, S. Ruiz-Moreno, J. R. D. Frejo, J. M. Maestre, and E. F. Camacho, “A fast implementation of coalitional model predictive controllers based on machine learning: Application to solar power plants,” *Engineering Applications of Artificial Intelligence*, vol. 118, p. 105666, 2023. DOI: 10.1016/j.engappai.2022.105666.
- [15] D. Li and S. Liu, “Structured light based high precision 3d measurement and workpiece pose estimation,” in *2019 Chinese Automation Congress (CAC)*, IEEE, 2019, pp. 669–674. DOI: 10.1109/CAC48633.2019.8996850.
- [16] R. Zaiabel, E. Dagan, J. Karni, and H. Ries, “An astigmatic corrected target-aligned heliostat for high concentration,” *Solar Energy Materials and Solar Cells*, vol. 37, no. 2, pp. 191–202, 1995. DOI: 10.1016/0927-0248(94)00206-1.

REVIEWER COMMENTS

Reviewer #2 (Remarks to the Author):

The authors have addressed the main questions posed in my review. Although it can be subject to further exploration in future works, I have still some concerns about the answer given to comment 2.1. The combination of environmental variations can be very large and many more images would be needed. The authors should have included in their study an estimation of the error made in terms of the deviation of the conditions under which they took the 4 images from other possible situations that can be encountered in daily practice. Anyway, this can be analysed in future works.

Reviewer #2 (Remarks on code availability):

The record is publicly accessible, but files are restricted to users with access

Reviewer #3 (Remarks to the Author):

The authors have well-revised the manuscript and implemented most of the corrections. Still, I would like to highlight a few issues that remain untouched after the first revision.

1. Optical properties still need to be included in the manuscript. It should specify the reflectivity, transmissivity, and absorptivity of both the reflector and absorber.

2. Table II in Supplementary material- Reflectivity mirror =1 needs to be more practical. In addition, the Reflectivity of target = ideal white does not indicate the exact values of an optical property for the absorber surface.

3. Since the geometry of the heliostat and absorber surface is excluded from the study, the authors should mention the parameters that affect the spread of the reflected images.

4. Reviewer Comment 3.4: No mention of heliostat is "aligned with the target" in the revised manuscript along with reference [16].

5. References [14], [15], [16] are much older literature for referencing aiming strategy and comparison. The authors can refer to recent literature <https://doi.org/10.1115/1.4053452> to specify the significance of the aiming strategy and image spread and shape.

6. Results of deflectometry are compared with prediction (ray tracing) based on the image shape. However, how the image spread or spot size is measured since there is no scale associated with Figure 1(c), Figure 3, and Figure 4.

Notes on revision made to manuscript NCOMMS-23-47231A-Z

The authors would like to thank the reviewers once again for their constructive comments and suggestions, which put the finishing touches to the manuscript. We now provide a very detailed background to our measurements, refer to recent literature and provide a more in depth discussion of the results with regard to measurement errors and environmental conditions. For more details, please see below for our individual responses. For the reviewers' convenience, we have highlighted our edits and replies from the first review in blue and from the second review in violet.

Response to the Reviewers

Reviewer 1

Reviewer Comment 1.1 — The authors have addressed the main questions posed in my review. Although it can be subject to further exploration in future works, I have still some concerns about the answer given to comment 2.1. The combination of environmental variations can be very large and many more images would be needed. The authors should have included in their study an estimation of the error made in terms of the deviation of the conditions under which they took the 4 images from other possible situations that can be encountered in daily practice. Anyway, this can be analysed in future works.

Reply: We agree, that environmental conditions affect the prediction accuracy as well as the data set size. We now mention this in the discussion section and we will investigate this in future work.

Reviewer Comment 1.2 — The record is publicly accessible, but files are restricted to users with access

Reply: We have uploaded the files in such a way that they automatically become freely accessible to everyone when the paper is published. We thought that this would also include early access for reviewers. If you want to have a look at the data before publication, we have attached the data as a zip file this time.

Reviewer 2

Reviewer Comment 2.1 — The authors have well-revised the manuscript and implemented most of the corrections. Still, I would like to highlight a few issues that remain untouched after the first revision.

1. Optical properties still need to be included in the manuscript. It should specify the reflectivity, transmissivity, and absorptivity of both the reflector and absorber. 2. Table II in Supplementary material- Reflectivity mirror =1 needs to be more practical. In addition, the Reflectivity of target = ideal white does not indicate the exact values of an optical property for the absorber surface.

Reply: The reflector is a heliostat using a solar mirror with a mean solar reflectivity of 94% (ISO9050, AM1.5). The glass surface has a waviness according to DIN EN 572-1/-2. For this study, no reflectivity measurement of the exact mirror used in this publication was conducted but we included a typical reflectivity for this kind of mirror in the supplementary.

The calibration target is coated with a 25 μm Polyvinylidene fluoride (PVDF) varnish in the defined color 9010. Literature shows absorption values for different white colors in between 0.2 and 0.4 [1]. A more precise determination of the solar absorptance of the calibration target was made possible by determining the solar reflectance of a sample of the coated sheet metal in a laboratory using the Perkin Lambda 950 photospectrometer. The reflectance values for the visible light spectrum were determined in 5 nm steps and weighted using the solar spectrum. The entire measurement curve is now provided in the supplementary.

While we believe the primary findings and conclusions of our research are clear without this detailed information, we understand that providing these optical properties can offer a more comprehensive understanding of our experimental setup and enhance the reproducibility of our results.

Reviewer Comment 2.2 — 3. Since the geometry of the heliostat and absorber surface is excluded from the study, the authors should mention the parameters that affect the spread of the reflected images.

Reply: Thank you again, we now go into more depth regarding the blurring of the focal spot according to distance, atmospheric losses, soiling and microscopic roughness of the mirror, Both in the main body of the text and in the supplementary.

Reviewer Comment 2.3 — 4. Reviewer Comment 3.4: No mention of heliostat is “aligned with the target” in the revised manuscript along with reference [16].

Reply: We mentioned it in the chapter *Field Test at a Concentrating Solar Power Plant*. However we forgot to highlight it in blue. We are sorry for this oversight. We now highlighted it in violet and added another mention in the *Sensitivity Analysis*.

Reviewer Comment 2.4 — 5. References [14], [15], [16] are much older literature for referencing aiming strategy and comparison. The authors can refer to recent literature <https://doi.org/10.1115/1.4053452> to specify the significance of the aiming strategy and image spread and shape.

Reply: Thank you. Citing a source from 2022 certainly highlights the relevance and topicality of this research. We added it to our introduction.

Reviewer Comment 2.5 — 6. Results of deflectometry are compared with prediction (ray tracing) based on the image shape. However, how the image spread or spot size is measured since there is no scale associated with Figure 1(c), Figure 3, and Figure 4.

Reply: We do not compare the image spread or the spot size; instead, we calculate the pixel-wise mean absolute error of the entire calibration target as described in Section *Learning NURBS*. Due to our formulation, the loss is independent of the image resolution and calibration target size. However, the images in Fig. 4 are zoomed in for better visualization of the details. Nevertheless, the loss is calculated based on the characteristics of the tower’s target given in the supplement. We acknowledge that this

point was not sufficiently emphasized, so we have revised the text accordingly and added a more detailed description of our loss function to the Methods section.

References

- [1] J. H. Henninger, “Solar absorptance and thermal emittance of some common spacecraft thermal-control coatings,” NASA Goddard Space Flight Center, NASA Reference Publication (RP) NASA-RP-1121, NAS 1.61:1121, REPT-84F0248, Apr. 1984, Document ID: 19840015630, Accession Number: 84N23698, Acquisition Source: Legacy CDMS, Project: RTOP 845-17-07. [Online]. Available: <https://ntrs.nasa.gov/citations/19840015630>.

REVIEWERS' COMMENTS

Reviewer #3 (Remarks to the Author):

The authors have sorted out all issues and implemented corrections in the main manuscript. The manuscript can be accepted for publication.